# Pregnancy with mixed connective tissue disease: Exploration of factors influencing live birth outcomes

**Tsukasa Yoshida**[1]*, **Jun Takeda**[1]*, **Sumire Ishii**[1], **Masakazu Matsushita**[2], **Naoto Tamura**[2], **Atsuo Itakura**[1]

**1** Department of Obstetrics and Gynecology, Juntendo University Faculty of Medicine, Tokyo, Japan,
**2** Department of Internal Medicine and Rheumatology, Juntendo University School of Medicine, Tokyo, Japan

* tsukatic0819@gmail.com (TY); jtakeda@juntendo.ac.jp (JT)

**Data Availability Statement:** All relevant data are within the manuscript and its Supporting information files.

## Abstract

Mixed connective tissue disease (MCTD) predominantly affects women in their reproductive age (30–40 years). This study is aimed to analyze a case series of MCTD-complicated pregnancies. The study design utilized a combined case-series and case–control approach. Pregnant women with MCTD were included and categorized into two groups: the live-birth and non-live birth (encompassing miscarriages at <12 weeks and stillbirths at $\geq$12 weeks) groups. Primary outcomes included delivery outcomes and factors associated with live births. A total of 57 pregnancies from 34 mothers (median age: 33.0 years) were included. Regarding delivery outcomes, the rates for live birth, miscarriage, and stillbirth were 64.9, 29.8, and 5.3%, respectively. Additionally, the respective rates of preterm delivery, fetal growth restriction (FGR), and small-for-gestational-age (SGA) were 18.9, 18.9, and 27.0%. Higher steroid usage (62.2 vs. 30.0%, $p = 0.02$) and lower prednisolone dosage in the live birth group (median dose: 7 vs. 10 mg, $p = 0.03$) were found to be significant factors contributing to live births. MCTD during pregnancy was associated with increased risks of miscarriage, stillbirth, preterm delivery, FGR, and SGA. Notably, low-dose steroid therapy was identified as a contributing factor to successful live births.

## Introduction

Mixed connective tissue disease (MCTD), first described by Sharp in 1972, is a connective tissue disorder (CTD) characterized by overlapping features of systemic lupus erythematosus (SLE), scleroderma, and polymyositis, accompanied by the presence of anti-U1-ribonucleoprotein antibodies [1]. Epidemiological studies report a prevalence of 3.8 per 100,000 individuals and an annual incidence of 2.1 per million, with pulmonary hypertension (PH) being the primary cause of mortality [2]. Moreover, MCTD has been reported to predominantly affect women in their reproductive age (30–40 years), raising significant concerns regarding pregnancy outcomes in these patients.

**Funding:** The author(s) received no specific funding for this work.

**Competing interests:** The authors have declared that no competing interests exist.

Studies have highlighted the association between MCTD and pregnancy, particularly on the complications of antiphospholipid syndrome and worsening PH during pregnancy in women with MCTD [3]. Furthermore, cases of complete heart block (CHB) have been reported in infants born to mothers with positive anti-U1-ribonucleoprotein antibodies despite negative anti-Sjögren's-syndrome(SS)-related antigen A (SS-A) and anti-SS-B antibodies [4].

Regarding the management of a pregnancy with MCTD, Radin et al. [5] reported that among 203 pregnancies from 96 mothers, cases that were given steroids and azathioprine demonstrated a higher live birth rate. Despite this, studies on MCTD-complicated pregnancies are limited, and a definitive consensus on pregnancy and delivery outcomes remains unclear. Additionally, although variations have been reported in the incidence and symptoms of CTD across different ethnicities [6], extensive data from Asian populations are lacking.

Given these circumstances, this study aimed to investigate a case series of MCTD-complicated pregnancies in Asian individuals, with the intention to provide valuable insights for the management of pregnancy and delivery in this specific cohort.

## Methods

The study employed a combined case-series and case–control design and was conducted from January 2001 to June 2023. This study design was selected considering the low prevalence of MCTD-complicated pregnancies. The case series design allows for the detailed collection and analysis of individual cases, whereas the case-control study is well-suited for examining exposures and identifying more generalizable results. Pregnant women with MCTD were included and categorized into two groups: the live-birth and non-live birth (encompassing miscarriages at <12 weeks and stillbirths at ≥12 weeks) groups. Miscarriages and stillbirths were classified based on gestational age, and all cases were confirmed via routine prenatal examinations. These classifications were verified through medical records. Additionally, imaging findings used in diagnosis were reviewed to maintain data accuracy.

MCTD diagnosis was confirmed based on the diagnostic criteria published in 1987 by Kasukawa et al., as revised in 1996 or 2004 by the representatives of the MCTD research committee of the Japanese Ministry of Health and Welfare [7], as well as a confirmatory diagnosis from a qualified rheumatologist. Similarly, SLE diagnosis was confirmed based on the American College of Rheumatology (ACR) criteria [8] and diagnosis from a qualified rheumatologist.

The inclusion criteria were: Asian pregnant women diagnosed with MCTD form the commencement of their pregnancy, and managed at Juntendo University, Tokyo, Japan. The exclusion criteria were cases with suspected MCTD and non-Asian women. The exclusion of non-Asian women was based on the study's objective to investigate a specific cohort where cultural and genetic factors, such as immune cell gene expression [9], might influence the outcomes. The primary outcomes of the study were delivery outcomes and factors associated with live births.

Univariate analysis was performed using the Mann–Whitney $U$ test and Fisher's exact test, whereas multivariate analysis utilized the classification and regression trees (CART) model. All statistical analyses were performed using R (version 3.5.0; R Foundation for Statistical Computing, Vienna, Austria) and EZR (Jichi Medical University Saitama Medical Center, Saitama, Japan) [10].

This study was approved by the Juntendo University School of Medicine Research Ethics Committee (registration number, E21-0070) and adhered with the principles outlined in the Declaration of Helsinki. This study adopted an opt-out consent approach for patient participation due to the study's retrospective and observational design. This implied study inclusion by

default unless individuals explicitly chose not to participate. The date on which the data were accessed for research purposes was August 1st, 2023. Patient information were retrieved from a secure and comprehensive database system, with approval from the Ethics Committee. To ensure accuracy, multiple obstetricians and rheumatologists verified all the data utilized in the study.

To ensure data consistency during the extended study period, we conducted regular data audits. Specifically, we confirmed that diagnostic criteria were standardized based on the guidelines available at the time of data collection, and treatment protocols were consistently aligned with these guidelines.

## Results

The study included 57 pregnancies from 34 mothers, with a median age of 33.0 years. Among them, 14.0% were multiparous, and 10.5% conceived through assisted reproductive treatment (Table 1). All patients in our study were cognizant and consented to their conception. Regarding MCTD management during pregnancy, 29 cases (50.9%) used prednisolone, and 4 (7.0%) used immunosuppressants (Table 1). In all four cases, tacrolimus was used. As for complications, two cases developed PH, with no cases of kidney dysfunction (Table 1). Eight cases also developed complications in SLE, although all were negative for anti-Smith antibodies and showed no severe organ involvement like lupus nephritis (Table 1). Although these cases fulfilled the American College of Rheumatology criteria for SLE, MCTD was diagnosed based on their clinical manifestations, such as scleroderma symptoms and myositis. All cases were confirmed to be U1-RNP positive. Additionally, one case was positive for cardiolipin antibodies, but no cases met the criteria for antiphospholipid-related syndromes. Eight cases were positive for anti SS-A antibodies, but none of the newborns developed congenital heart block (Table 1).

Regarding delivery outcomes, rates for live birth and stillbirth were 64.9 and 5.3%, respectively (Table 1). All 20 cases of miscarriage and stillbirth were identified during routine

**Table 1. Patient background.**

| Variables | Overall (n = 57) |
|---|---|
| Age[†], years | 33.0 (24.0–36.0) |
| Multiparous[‡] | 8 (14.0) |
| Fertility treatment[‡] | 6 (10.5) |
| Pulmonary hypertension[‡] | 2 (3.5) |
| SLE complication | 8 (14.0) |
| Anti SS-A antibodies[‡] | 12 (21.1) |
| Prednisolone usage[‡] | 29 (50.9) |
| Prednisolone[†], mg/day | 7.25 (3.0–12.0) |
| Immunosuppressant[‡] | 4 (7.0) |
| Heparin[‡] | 4 (7.0) |
| Low dose aspirin[‡] | 3 (5.3) |
| Live birth[‡] | 37 (64.9) |
| Miscarriage[‡] | 17 (29.8) |
| Stillbirth[‡] | 3 (5.3) |

[†]Data are presented as median (range) for continuous variables.

[‡]Data are presented as number (%) for categorical variables,

Abbreviations: SLE, Systemic Lupus Erythematosus; anti-SS-A, anti-Sjögren's-syndrome-related antigen A.

**Table 2. Univariate analysis between the live-birth and non-live birth groups.**

| Variables | Live birth (n = 37) | Non-live birth (n = 20) | p-value |
|---|---|---|---|
| Age†, years | 33.0 (24.0–36.0) | 33.0 (28.0–35.0) | 1.00 |
| Multiparous‡ | 6 (18.8) | 2 (22.2) | 1.00 |
| Fertility treatment‡ | 6 (16.2) | 0 (0.0) | 0.09 |
| SLE complication | 4 (20.0) | 4 (36.4) | 0.405 |
| Anti-SS-A antibodies‡ | 11 (37.9) | 1 (16.7) | 0.64 |
| Prednisolone usage‡ | 23 (62.2) | 6 (30.0) | 0.02* |
| Prednisolone†, mg/day | 7 (3–11) | 10 (10–12) | 0.03* |
| Other immunosuppressant‡ | 3 (10.0) | 1 (16.7) | 0.54 |
| Heparin‡ | 3 (33.3) | 1 (25.0) | 1.00 |
| Low dose aspirin‡ | 3 (17.6) | 0 (0.0) | 1.00 |

†Data are presented as median (range) for continuous variables.

‡Data are presented as number (%) for categorical variables.

* $p < 0.05$.

Abbreviations: SLE, Systemic Lupus Erythematosus, anti-SS-A, anti-Sjögren's syndrome-related antigen A.

prenatal examinations. Notably, no terminations of pregnancy were medically necessary due to worsening MCTD severity.

No significant differences were observed in age, SLE complications, and anti-SS-A antibodies between the live-birth and non-live birth groups (Table 2). However, the live birth group exhibited a significantly higher rate of prednisolone use (62.2 vs. 30.0%, p = 0.02). In contrast, when dosage of steroids was compared among the patients using steroids, the non-live birth group had a statistically significantly higher median dose of prednisolone compared to that of the live birth group (10 vs. 7 mg/day, p = 0.03). (Table 2). Conversely, no significant differences were observed in other immunosuppressant, low-dose aspirin, and heparin uses between the two groups (Table 2). Interestingly, upon the incorporation of the significant parameters to the CART model, prednisolone use was identified as the most significant factor associated with live births (50.0 vs. 79.3%, $p = 0.02$), followed by prednisolone dose (<10 mg/day; 70.0 vs 100.0%, $p = 0.04$) (Fig 1).

Further analysis of delivery outcomes revealed a preterm delivery rate of 18.9% within live birth group (Table 3). Similarly, the rates of fetal growth restriction (FGR) small for gestational age (SGA) were 18.9 and 27%, respectively (Table 3). Fortunately, no cases of maternal mortality were reported (Table 3). In addition, Furthermore, no significant differences were observed in delivery outcomes between groups receiving and not receiving prednisolone (Table 3).

## Discussion

Our study demonstrated that the delivery outcomes in pregnant women with MCTD resulted in a live birth rate of 64.9% and stillbirth rate of 5.3%. Regarding factors contributing to live birth, significantly higher prednisolone usage in the live birth group (62.2 vs. 30.0%, $p = 0.02$) and higher dosage of prednisolone in the non-live birth group (median dose: 7 vs. 10 mg, $p = 0.03$) were identified as significant factors. Multivariate analysis via the CART model further identified prednisolone usage as the most significant factor, followed by prednisolone dosage <10 mg/day. Regarding the specific delivery outcomes, rate of preterm birth, FGR, and SGA were 18.9, 18.9, and 27.0%, respectively.

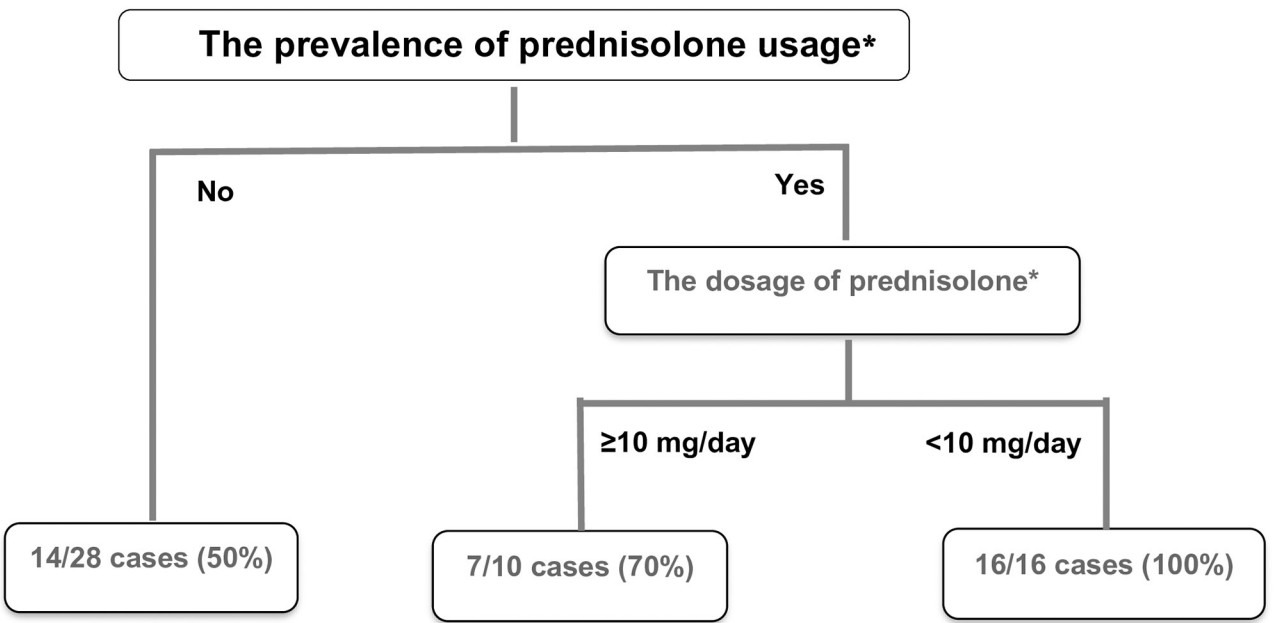

**Fig 1. Multivariate analysis of factors associated with live birth using the classification and regression trees model.** Prednisolone usage was identified as the most significant factor (50.0 vs. 79.3%, $p = 0.02$), followed by prednisolone dosage (<10 mg/day; 70.0 vs 100.0%, $p = 0.04$). *$p<0.05$: Indicates a statistically significant difference.

A previous study reported a live birth rate of 72% and a stillbirth rate of 8.9% in MCTD pregnancies [5], which was consistent with our findings. Notably, the stillbirth rate reported in the present study was significantly higher compared to the national prevalence of 0.3% and Asian prevalence of 2.0% [11]. Compared to other CTDs, stillbirth rates in MCTD pregnancies

**Table 3. Delivery outcomes in pregnant women with MCTD.**

| Variables | | Overall (n = 37) | | Prednisolone (+) (n = 23) | | Prednisolone (-) (n = 14) | | *p*-value |
|---|---|---|---|---|---|---|---|---|
| Gestational week[†], week | | 38.0 (35.0–40.0) | | 38.0 (36.0–40.0) | | 38.0 (35.0–40.0) | | 0.91 |
| Preterm birth[‡], | | 7 (18.9) | | 4 (17.4) | | 3 (21.4) | | 1.00 |
| Birthweight[†], g | SGA[‡] | 2542 (1820–3678) | 10 (27.0) | 2510 (2052–3678) | 2 (8.7) | 2544 (1820–3362) | 8 (57.1) | 0.43 |
| | AGA[‡] | | 22 (59.5) | | 9 (39.1) | | 13 (92.9) | (0.27) |
| | LGA[‡] | | 3 (8.1) | | 2 (8.7) | | 1 (7.1) | |
| Labor duration [†], h | | 6.5 (2.0–18.5) | | 9.0 (8.0–9.0) | | 9.0 (8.0–9.0) | | 0.87- |
| Bleeding volume[†], mL | | 472 (80.0–2405.0) | | 472 (218–800) | | 490 (80–2405) | | 0.93 |
| Mode of delivery; CS | | 13 (35.1) | | 11 (47.8) | | 2 (14.2) | | 0.07 |
| Fetal growth restriction[‡] | | 7 (18.9) | | 5 (21.7) | | 2 (14.3) | | 0.69 |
| Hypertensive Disorder of Pregnancy[‡] | | 2 (5.4) | | 1 (4.3) | | 1 (7.1) | | 1.00 |
| Gestational Diabetes Mellitus[‡] | | 2 (5.4) | | 2 (8.7) | | 0 (0.0) | | 0.52 |
| Placenta previa[‡] | | 0 (0.0) | | 0 (0.0) | | 0 (0.0) | | 1.00 |

[†]Data are presented as median (range) for continuous variables.

[‡]Data are presented as number (%) for categorical variables.

* $p<0.05$.

Abbreviations: AGA, appropriate for gestational age; CS, cesarian section; LGA, Large for gestational age; SGA, small-for-gestational-age.

surpass those observed in SLE pregnancies, which has a reported live birth rate of 90% [12]. As such, the American College of Obstetricians and Gynecologists Committee recommends weekly antenatal fetal surveillance for pregnant women with SLE at 32 weeks age of gestation [13]. However, they acknowledge the lack of sufficient evidence for similar recommendations in MCTD pregnancies. Our findings suggest the need to reevaluate this recommendation for MCTD.

Although relatively higher MCTD activity was expected in steroid users, our study observed lower stillbirth rates in patients who used steroids. Recent studies suggest that the effects of prednisolone on the immune system may vary depending on the individual's immune status [14]. As highlighted in previous reviews, although suppression of peri-conception inflammation might impair immune function in some women, it is possible that in women predisposed to excessive inflammation and immunity, such as those with MCTD, prednisolone could help shift the balance of the immune response towards M2 macrophages, tolerogenic dendritic cells, and a stronger regulatory T cell response, thereby promoting a more favorable pregnancy outcome [14]. Steroid use in patients with unexplained recurrent pregnancy loss has been reported to downregulate natural killer cells in the uterine endometrium, thereby reducing miscarriage rates [15]. It is therefore possible that a similar mechanism contributed to the prevention of stillbirth with steroid use in MCTD. This aligns with previous studies that has shown that a combination of azathioprine and steroids significantly improves pregnancy outcomes in MCTD [5]. Similarly, prednisone has been observed to effectively manage disease activity and contribute to better delivery outcomes in other CTDs, such as SLE [16]. This is consistent with the European Alliance of Associations for Rheumatology (EULAR) points to consider for the use of antirheumatic drugs during pregnancy, which recommend continuation of certain corticosteroids like prednisolone for effective management of disease activity [17].

Despite the advantage of steroid use, our research also demonstrated that higher doses of steroid were associated with unfavorable outcomes during pregnancy. Studies on SLE have shown that steroid doses >6.5 mg/day were associated with adverse pregnancy outcomes, particularly increasing preterm birth incidence with doses >10.0 mg/day [18]. Our results were consistent with those of previous studies, showing that doses >10 mg/day resulted in unfavorable live birth outcomes. In line with the EULAR recommendations, our findings support minimizing the use of high-dose steroids during pregnancy to reduce the risk of adverse outcomes [19]. The guidelines, along with our study, emphasize the importance of combination therapy with other immunosuppressive agents to achieve optimal disease control while minimizing steroid dosage [19]. Studies have also reported that high-dose steroids can increase the risk of placenta accreta formation and uterine myometrial fibrosis due to reduced estrogenic effect and endometrial thinning, potentially causing atonic bleeding and substantial blood loss [20, 21]. Thus, minimizing the total steroid dose whenever possible is crucial to ensure optimal outcomes for both the mother and fetus.

Regarding specific delivery outcomes, our study reported a significantly higher preterm birth rate compared to the national average of 5.0% and other Asian rates [22, 23]. Similarly, previous studies have reported high rates of fetal complications in MCTD-complicated pregnancies, with prematurity observed in 48.1% and intrauterine growth restriction in 38.3% of cases [24]. While the reasons for increased FGR incidence in MCTD-complicated pregnancies remain unclear, a mechanism similar to the increased incidence of FGR/SGA (approximately 10–30%) in SLE may be possible. Particularly, the increased occurrence of elevated blood pressure and increased blood glucose levels were shown to be associated with worsening SLE symptoms [25, 26]. Unfortunately, comorbidities such as preeclampsia and gestational diabetes were not investigated in this study, precluding a more comprehensive evaluation of risk

factors for FGR/SGA. Nevertheless, the presence of CTDs has been strongly hinted to influence the development of FGR/SGA. Interestingly, according to the Developmental Origins of Health and Disease concept, a potential link between CTDs during pregnancy and fetal growth could increase the risk of developing non-communicable diseases in the infant, including diabetes, cancer, and cardiovascular conditions [27]. This perspective warrants further discussion, as this is crucial to understanding the association between CTDs and risk factors in specific delivery outcomes.

Despite the insights offered in this study, certain limitations should be acknowledged. First, there was limited generalizability of findings due to the study's retrospective, single-center design. This design introduces limitations such as selection bias and information bias, as data were collected from past medical records. The study results could have been influenced by changes in clinical practices and variations in data recording over time. Second, steroid use solely relied on clinical judgment, resulting in the lack of a standardized steroid administration protocol. This reliance on clinical judgment introduces potential bias, as the decisions regarding dosage and administration may vary depending on the physician's experience and knowledge, which could have influenced the pregnancy outcomes observed in this study. Third, the study only focused on the maximum dosage of steroids. Future studies might benefit from including the cumulative dosage to further elucidate the association between steroid use and MCTD-complicated pregnancy. Additionally, multicenter studies are recommended to enhance the generalizability of the findings by incorporating a more diverse patient population. The development and implementation of standardized protocols for steroid use in MCTD-complicated pregnancies are also crucial. Such protocols would help to minimize variability in treatment and provide more consistent data on the effects of steroid use.

## Conclusions

In conclusion, this study demonstrated that MCTD significantly impacts pregnancy outcomes, resulting in increased rates of miscarriage, stillbirth, preterm birth, FGR, and SGA. In addition, the use of low-dose steroids was associated with improved live birth rates in pregnant women with MCTD.

## Supporting information

**S1 Data.**
(XLSX)

## Author Contributions

**Conceptualization:** Tsukasa Yoshida.

**Data curation:** Tsukasa Yoshida, Sumire Ishii.

**Formal analysis:** Tsukasa Yoshida.

**Investigation:** Tsukasa Yoshida, Sumire Ishii.

**Methodology:** Tsukasa Yoshida, Jun Takeda, Sumire Ishii, Masakazu Matsushita, Naoto Tamura.

**Project administration:** Tsukasa Yoshida, Jun Takeda, Atsuo Itakura.

**Resources:** Tsukasa Yoshida, Sumire Ishii.

**Software:** Tsukasa Yoshida.

**Supervision:** Jun Takeda, Masakazu Matsushita, Naoto Tamura, Atsuo Itakura.

**Validation:** Tsukasa Yoshida, Jun Takeda.

**Visualization:** Tsukasa Yoshida.

**Writing – original draft:** Tsukasa Yoshida.

**Writing – review & editing:** Tsukasa Yoshida, Jun Takeda, Masakazu Matsushita, Naoto Tamura, Atsuo Itakura.

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
