## [Decision Letter · Decision Letter 0]

7 Aug 2024

PONE-D-24-14916Pregnancy with mixed connective tissue disease: Exploration of factors influencing live birth outcomesPLOS ONE

Dear Dr. Yoshida,

Thank you for submitting your manuscript to PLOS ONE. After careful consideration, we feel that it has merit but does not fully meet PLOS ONE’s publication criteria as it currently stands. Therefore, we invite you to submit a revised version of the manuscript that addresses the points raised during the review process.

Please note that we have only been able to secure a single reviewer to assess your manuscript. We are issuing a decision on your manuscript at this point to prevent further delays in the evaluation of your manuscript. Please be aware that the editor who handles your revised manuscript might find it necessary to invite additional reviewers to assess this work once the revised manuscript is submitted. However, we will aim to proceed on the basis of this single review if possible.

We look forward to receiving your revised manuscript.

Kind regards,

Annesha Sil, Ph.D.

Associate Editor

PLOS ONE

Journal Requirements:

Reviewers' comments:

Reviewer's Responses to Questions

**Comments to the Author**

1. Is the manuscript technically sound, and do the data support the conclusions?

Reviewer #1: Yes

2. Has the statistical analysis been performed appropriately and rigorously? 

Reviewer #1: Yes

3. Have the authors made all data underlying the findings in their manuscript fully available?

Reviewer #1: Yes

4. Is the manuscript presented in an intelligible fashion and written in standard English?

Reviewer #1: Yes

5. Review Comments to the Author

Reviewer #1: This study aimed to investigate a case series of MCTD-complicated pregnancies in Asian individuals.

I have several comments:

1) The combined case-series and case-control design allows for a detailed examination of both individual cases and broader patterns, which is a commendable approach. The rationale for choosing this design is not explicitly stated. Clarifying why this design was preferred over others would strengthen the methodological justification.

2) No information is provided on how data consistency was maintained over this long period. The potential for changes in diagnostic criteria, treatment protocols, or data recording practices over time should be addressed.

3) The exclusion of non-Asian women limits the generalizability of the findings. The justification for this exclusion should be elaborated. Additionally, the handling of suspected MCTD cases should be clarified—what criteria were used to exclude them, and how were borderline cases managed? The live-birth and non-live birth groups are well-defined, but further explanation on how miscarriages and stillbirths were confirmed and categorized would enhance clarity.

4) Describe the data collection process in more detail. How were patient records accessed and verified? How was data consistency ensured over the extended study period?

5) In the result section, there is no sufficient information on immunology (autoantibodies, including APL, anti-DNAds, anti-SM, anti-SSA/Anti-SSB), type of immunossupression before pregnancy, doses of corticosteroids before and during pregnancy and the clinical manfiestations of the MCTD, including PH, which is referenced as being one of the causes for pregnancy morbidity.

6) "However, the live birth group exhibited a significantly higher rate of prednisolone use (62.2% vs. 30.0%, p=0.02), while the non-live birth group showed a greater prevalence for higher doses of prednisolone (median dose: 7 vs. 10 mg/day, p=0.03) (Table 2)". This sentence is very confusing and the data the authors present is insuficient to get this conclusion. Does this mean that corticosteroid use is associated with favorable outcomes? But doses higher than 10 mg/day were associated with unfavourable outcomes?

7) "Conversely, no significant differences were observed in other immunosuppressant, low-dose aspirin, and heparin uses between the two groups (Table 2)." There is no mention of the type of immunosuppressive treatment used and this is important.

8)The discussion lacks depth in comparing the findings with a broader range of studies. It relies heavily on a few sources, which may limit the scope of the comparison.

9) There is no mention of these important guidelines:

Götestam Skorpen C, Hoeltzenbein M, Tincani A, et alThe EULAR points to consider for use of antirheumatic drugs before pregnancy, and during pregnancy and lactationAnnals of the Rheumatic Diseases 2016;75:795-810.

Andreoli L, Bertsias GK, Agmon-Levin N, et alEULAR recommendations for women's health and the management of family planning, assisted reproduction, pregnancy and menopause in patients with systemic lupus erythematosus and/or antiphospholipid syndromeAnnals of the Rheumatic Diseases 2017;76:476-485.

10) The discussion provides a detailed analysis of the impact of steroid use on pregnancy outcomes, highlighting both positive effects (improved live birth rates) and negative effects (higher doses associated with unfavorable outcomes). The explanation of the mechanisms by which steroids might improve outcomes (e.g., suppression of autoantibodies, downregulation of natural killer cells) is somewhat speculative and could benefit from more robust evidence or references.

11) The limitations could be expanded to discuss more potential biases and confounding factors. The reliance on clinical judgment for steroid use and the retrospective, single-center design are mentioned, but their implications on the results could be explored further.

12) The discussion on future research is brief. It should include more detailed recommendations for how future studies can build on these findings, including suggestions for multicenter studies or standardized protocols for steroid use.

6. PLOS authors have the option to publish the peer review history of their article (what does this mean?). If published, this will include your full peer review and any attached files.

Reviewer #1: **Yes: **Cristiana Sieiro Santos

---

## [Author Response · Author response to Decision Letter 0]

16 Sep 2024

Re: Manuscript ID: PONE-D-24-14916

September 16, 2024

Emily Chenette

Editor in Chief, PLoS One

Thank you for giving us the opportunity to submit a revised draft of our manuscript titled, “Pregnancy with mixed connective tissue disease: Exploration of factors influencing live birth outcomes” to PLoS One.

We appreciate the time and effort that you and the reviewers have dedicated to providing your valuable feedback on our manuscript. We are grateful for their insightful comments. We have been able to incorporate changes to reflect most of the suggestions provided by the reviewers, and we believe these changes have greatly improved the quality of our manuscript. We hope that with these changes, the manuscript meets with your and the reviewers’ approval.

Please find attached herewith our point-by-point responses to the reviewers’ comments. Revisions in the manuscript have been indicated with yellow highlighted text. The page and line numbers have been transcribed in accordance with the tracked version of the manuscript.

Many parts of the manuscript have been carefully rewritten to enhance the flow and clarity of some previously ambiguous text. We hope that these revisions and our accompanying responses would bring our manuscript closer to publication in the PLOS ONE.

Point-by-point response to the reviewers' comments

We would like to thank the Reviewer for spending your valuable time to review our manuscript and the constructive critique to improve our manuscript. The manuscript has benefited immensely from the suggestions provided. We have made every effort to address the issues raised and to respond to all the comments and suggestions and have made the necessary changes in accordance with the suggestions and believe that the changes made in this round of revision would satisfactorily address the concerns raised. Please find our response to each item below.

1) The combined case-series and case-control design allows for a detailed examination of both individual cases and broader patterns, which is a commendable approach. The rationale for choosing this design is not explicitly stated. Clarifying why this design was preferred over others would strengthen the methodological justification.

Response： Thank you for your insightful comment. We chose this study design considering the low prevalence of MCTD-complicated pregnancies. The case series design allows for the detailed collection and analysis of individual cases, whereas the case-control study is well-suited for examining exposures and identifying broader patterns and trends. By comprehensively evaluating both distinct cases and more generalizable results, we aim to deepen the understanding of MCTD-complicated pregnancies. We have added this explanation to the manuscript in method part.

Line 61-64：”This study design was selected considering the low prevalence of MCTD-complicated pregnancies. The case series design allows for the detailed collection and analysis of individual cases, whereas the case-control study is well-suited for examining exposures and identifying more generalizable results.”

2) No information is provided on how data consistency was maintained over this long period. The potential for changes in diagnostic criteria, treatment protocols, or data recording practices over time should be addressed.

Response: Thank you for highlighting the importance of data consistency over the extended study period. We took several measures to ensure data consistency, including regular data audits, standardization of diagnostic criteria based on the guidelines available at the time of data collection, and consistent alignment of treatment protocols with these guidelines. We have now added these details to the revised manuscript.

Line 98-101：“To ensure data consistency during the extended study period, we conducted regular data audits. Specifically, we confirmed that diagnostic criteria were standardized based on the guidelines available at the time of data collection, and treatment protocols were consistently aligned with these guidelines.”

3) The exclusion of non-Asian women limits the generalizability of the findings. The justification for this exclusion should be elaborated. Additionally, the handling of suspected MCTD cases should be clarified—what criteria were used to exclude them, and how were borderline cases managed? The live-birth and non-live birth groups are well-defined, but further explanation on how miscarriages and stillbirths were confirmed and categorized would enhance clarity.

Response: 

Thank you for your valuable feedback. The exclusion of non-Asian women was based on the study’s objective to investigate a specific population where cultural and genetic factors, such as immune cell gene expression[9], might influence the outcomes. 

The exclusion criteria for suspected MCTD cases were based on the diagnostic criteria published in 1987 by Kasukawa et al., as revised in 1996 and 2004 by the representatives of the MCTD Research Committee of the Japanese Ministry of Health and Welfare. Additionally, all cases were reviewed by rheumatology specialists to ensure there were no discrepancies in the diagnosis and treatment of MCTD.

Miscarriages and stillbirths were classified based on gestational age, and all cases were confirmed through routine prenatal examinations. These classifications were verified through medical records, and imaging findings used in diagnosis were also reviewed to maintain data accuracy. We have added these clarifications to the manuscript.

Line 67-70: “Miscarriages and stillbirths were classified based on gestational age, and all cases were confirmed via routine prenatal examinations. These classifications were verified through medical records. Additionally, imaging findings used in diagnosis were reviewed to maintain data accuracy.”

Line 79-82: “The exclusion of non-Asian women was based on the study’s objective to investigate a specific cohort where cultural and genetic factors, such as immune cell gene expression[9], might influence the outcomes.”

Reference: 9, Yazar S, Alquicira-Hernandez J, Wing K, Senabouth A, Gordon MG, Andersen S, et al. Single-cell eQTL mapping identifies cell type-specific genetic control of autoimmune disease. Science. 2022;376: eabf3041. doi:10.1126/science.abf3041

4) Describe the data collection process in more detail. How were patient records accessed and verified? How was data consistency ensured over the extended study period?

Response: Thank you for your comments regarding the data collection process. We retrieved patient records from a secure and comprehensive database system, with all data accessed and confirmed after receiving approval from the relevant ethics committee. The data used in the study were double-checked by multiple obstetricians and rheumatologists to ensure accuracy. 

To maintain data consistency over the long study period, regular audits and data cleaning were conducted, and diagnostic criteria and treatment protocols were standardized according to the latest guidelines. (as also noted in our response to question 2).

We have now included these details in the manuscript.

Line 95-97: “Patient information were retrieved from a secure and comprehensive database system, with approval from the Ethics Committee. To ensure accuracy, multiple obstetricians and rheumatologists verified all the data utilized in the study. “

Line 98-101：“To ensure data consistency during the extended study period, we conducted regular data audits. Specifically, we confirmed that diagnostic criteria were standardized based on the guidelines available at the time of data collection, and treatment protocols were consistently aligned with these guidelines.” (as also noted in our response to question 2).

5) In the result section, there is no sufficient information on immunology (autoantibodies, including APL, anti-DNAds, anti-SM, anti-SSA/Anti-SSB), type of immunossupression before pregnancy, doses of corticosteroids before and during pregnancy and the clinical manfiestations of the MCTD, including PH, which is referenced as being one of the causes for pregnancy morbidity.

Response: Thank you for your feedback. We have added more detailed information on autoantibodies, immunosuppressive treatment, corticosteroid dosage before and during pregnancy, and the clinical manifestations of MCTD. Specifically, all MCTD cases were confirmed to be U1-RNP positive. Among the SLE-complicated cases, 8 were positive for ds-DNA antibodies, though none had anti-Sm antibodies or severe organ damage such as lupus nephritis, and no cases of kidney failure were observed. Additionally, one case was positive for cardiolipin antibodies, but no cases met the criteria for APS-related syndromes. Eight cases were positive for anti-SSA antibodies, but none of the newborns developed congenital heart block. Regarding immunosuppressive treatment, four cases involved the use of tacrolimus. There were no adjustments in prednisolone dosage during pregnancy, but two cases had increased doses postpartum. The clinical manifestations of MCTD included pulmonary hypertension in two cases, with no reports of kidney dysfunction. We have incorporated these details into the manuscript.

Line 107-118: “Regarding MCTD management during pregnancy, 29 cases (50.9%) used prednisolone, and 4 (7.0%) used immunosuppressants (Table 1). In all four cases, tacrolimus was used. As for complications, two cases developed PH, with no cases of kidney dysfunction (Table 1). Eight cases also developed complications in SLE, although all were negative for anti-Smith antibodies and showed no severe organ involvement like lupus nephritis (Table 1). Although these cases fulfilled the American College of Rheumatology criteria for SLE, MCTD was diagnosed based on their clinical manifestations, such as scleroderma symptoms and myositis. All cases were confirmed to be U1-RNP positive. Additionally, one case was positive for cardiolipin antibodies, but no cases met the criteria for antiphospholipid-related syndromes. Eight cases were positive for anti SS-A antibodies, but none of the newborns developed congenital heart block (Table 1).”

6) "However, the live birth group exhibited a significantly higher rate of prednisolone use (62.2% vs. 30.0%, p=0.02), while the non-live birth group showed a greater prevalence for higher doses of prednisolone (median dose: 7 vs. 10 mg/day, p=0.03) (Table 2)". This sentence is very confusing and the data the authors present is insuficient to get this conclusion. Does this mean that corticosteroid use is associated with favorable outcomes? But doses higher than 10 mg/day were associated with unfavourable outcomes?

Response: Thank you for pointing out the potential confusion in the original phrasing. We have revised the text in the Results section to clarify our findings. Specifically, we now state that “the live birth group exhibited a significantly higher rate of prednisolone use (62.2% vs. 30.0%, p=0.02). In contrast, when focusing on the patients who used steroids, the non-live birth group had a statistically significantly higher median dose of prednisolone (7 vs. 10 mg/day, p=0.03).” This revision should clarify that while prednisolone use was more common in the live birth group, higher doses were more prevalent in the non-live birth group. We have also discussed this in the Discussion section.

Line 131-136: “However, the live birth group exhibited a significantly higher rate of prednisolone use (62.2 vs. 30.0%, p=0.02). In contrast, when dosage of steroids was compared among the patients using steroids, the non-live birth group had a statistically significantly higher median dose of prednisolone compared to that of the live birth group (10 vs. 7 mg/day, p=0.03). (Table 2).”

7) "Conversely, no significant differences were observed in other immunosuppressant, low-dose aspirin, and heparin uses between the two groups (Table 2)." There is no mention of the type of immunosuppressive treatment used and this is important.

Response：Thank you for your valuable feedback. As mentioned above, the only immunosuppressants used by patients during the study period was tacrolimus. The details have been provided in the earlier section. Additionally, as shown in Table 2, there was no statistically significant difference in the usage rates between the two groups.

We have added the following sentence: Line 111-112: “In all four cases, tacrolimus was used.”

8) The discussion lacks depth in comparing the findings with a broader range of studies. It relies heavily on a few sources, which may limit the scope of the comparison.

Response: Thank you for your valuable feedback. We appreciate your suggestion to broaden the scope of our discussion by comparing our findings with a wider range of studies. In response, we have added additional references, including both case series and systematic literature reviews, to provide a more comprehensive comparison of our results.

Specifically, we have included references that highlight the high incidence of fetal complications in pregnancies complicated by MCTD, such as prematurity (48.1%) and intrauterine growth restriction (38.3%). This information has been integrated into the discussion to provide a broader context for our findings and to strengthen the overall comparison with other studies.

Additionally, we would like to express our gratitude for the comments provided in points No.9 and 10. These suggestions have allowed us to further refine our discussion, making it more nuanced and comprehensive.

We believe that these additions enhance the depth of our discussion and provide a more nuanced understanding of the challenges associated with MCTD-complicated pregnancies.

Thank you again for your thoughtful comments, which have contributed to improving the quality of our manuscript.

Line 227-235: “Regarding specific delivery outcomes, our study reported a significantly higher preterm birth rate compared to the national average of 5.0% and other Asian rates [22, 23]. Similarly, previous studies have reported high rates of fetal complications in MCTD-complicated pregnancies, with prematurity observed in 48.1% and intrauterine growth restriction in 38.3% of cases [24]. Although the reasons for increased FGR incidence in MCTD-complicated pregnancies remain unclear, a mechanism similar to the increased incidence of FGR/SGA (approximately 10–30%) in SLE may be possible. Particularly, the increased occurrence of elevated blood pressure and increased blood [25, 26].”

Reference：

24, Tardif M-L, Mahone M. Mixed connective tissue disease in pregnancy: A case series and systematic literature review. Obstet Med. 2019;12: 31–37.

9) There is no mention of these important guidelines:

Götestam Skorpen C, Hoeltzenbein M, Tincani A, et alThe EULAR points to consider for use of antirheumatic drugs before pregnancy, and during pregnancy and lactationAnnals of the Rheumatic Diseases 2016;75:795-810.

Andreoli L, Bertsias GK, Agmon-Levin N, et alEULAR recommendations for women's health and the management of family planning, assisted reproduction, pregnancy and menopause in patients with systemic lupus erythematosus and/or antiphospholipid syndromeAnnals of the Rheumatic Diseases 2017;76:476-485.

Response：Thank you for your valuable feedback. We have revised the Discussion section to incorporate guidelines from The European Alliance of Associations for Rheumatology (EULAR) Points to Consider for Use of Antirheumatic Drugs Before Pregnancy, and During Pregnancy and Lactation [17] as well as the EULAR Recommendations for Women’s Health and the Management of Family Planning, Assisted Reproduction, Pregnancy, and Menopause in Patients with Systemic Lupus Erythematosus (SLE) and/or Antiphospholipid Syndrome (APS) [19]. Specifically, we have referenced these guidelines to support the use of antirheumatic drugs like prednisolone during pregnancy, emphasizing the importance of continuing certain corticosteroids to effectively manage disease activity while minimizing risks to both the mother and fetus [17]. Additionally, we aligned our findings with the EULAR recommendations, which advise minimizing the use of high-dose steroids during pregnancy to reduce the risk of adverse outcomes [19].

---

## [Decision Letter · Decision Letter 1]

18 Oct 2024

Pregnancy with mixed connective tissue disease: Exploration of factors influencing live birth outcomes

PONE-D-24-14916R1

Dear Dr. Yoshida,

We’re pleased to inform you that your manuscript has been judged scientifically suitable for publication and will be formally accepted for publication once it meets all outstanding technical requirements.

Kind regards,

Linglin Xie

Academic Editor

PLOS ONE

Additional Editor Comments (optional):

Reviewers' comments:

Reviewer's Responses to Questions

**Comments to the Author**

1. If the authors have adequately addressed your comments raised in a previous round of review and you feel that this manuscript is now acceptable for publication, you may indicate that here to bypass the “Comments to the Author” section, enter your conflict of interest statement in the “Confidential to Editor” section, and submit your "Accept" recommendation.

Reviewer #1: (No Response)

2. Is the manuscript technically sound, and do the data support the conclusions?

Reviewer #1: Yes

3. Has the statistical analysis been performed appropriately and rigorously? 

Reviewer #1: Yes

4. Have the authors made all data underlying the findings in their manuscript fully available?

Reviewer #1: Yes

5. Is the manuscript presented in an intelligible fashion and written in standard English?

Reviewer #1: Yes

6. Review Comments to the Author

Reviewer #1: (No Response)

7. PLOS authors have the option to publish the peer review history of their article (what does this mean?). If published, this will include your full peer review and any attached files.

Reviewer #1: No

---

## [Editor Report · Acceptance letter]

22 Oct 2024

PONE-D-24-14916R1 

PLOS ONE

Dear Dr. Yoshida, 

I'm pleased to inform you that your manuscript has been deemed suitable for publication in PLOS ONE. Congratulations! Your manuscript is now being handed over to our production team.

Kind regards, 

on behalf of

Dr. Linglin Xie 

Academic Editor

PLOS ONE